# Abelard and Other Twelfth-Century Thinkers on Social Constructions

## Andrew W. Arlig

Brooklyn College, The City University of New York, Brooklyn, NY 11210, USA; aarlig@brooklyn.cuny.edu

**Abstract:** This article aims to supplement our understanding of later developments within European universities, that is, Scholastic thought, by attending to how certain pre-Scholastics, namely, Peter Abelard and other twelfth-century philosophers, thought about artifacts and social constructions more generally. It focuses on the treatment of artifacts that can be cobbled together out of Abelard's *Dialectica*. The article argues that Abelard attempts to sharply distinguish the world of things from the world of human-made objects. This is most apparent in his treatment of creation and human acts of making. Yet there are places in his thought where we see some hesitancy. Many of Abelard's peers seem to have drawn on the reasons why Abelard hesitates, and they blur the line between human-made objects such as houses and substances such as rocks and humans. Others seem to go further than Abelard—perhaps inspired by some of the thoughts that Abelard also entertains about social constructions such as days and speeches—and assert that even houses are merely convenient fictions.

**Keywords:** Abelard; twelfth-century philosophy; substance; whole; part; change; persistence; thing (res); fictionalism





## 1. Introduction

In Scholastic thought—by which I mean the philosophy largely based on a close reading of Aristotle that is found in the major European universities in the thirteenth and fourteenth centuries (and well beyond)—it is standard to allocate artifacts and other socially constructed entities to the non-substantial. They are accidental unities of some sort or other, even if their structure is quite complex, and they move together as a unified whole in virtue of being welded, glued, or nailed together. This, of course, does not settle all the questions that one might have about the ontology of socially constructed objects; just read the remaining contributions of this special issue. However, one of the most intriguing features of twelfth-century Latin philosophy is the fact that the line between substances and artifacts is often not so sharp. I will demonstrate this thesis by surveying one of Peter Abelard's more sustained reflections about the nature, identity, and persistence of artifacts such as houses and other potentially socially constructed items such as temporal wholes and numbers. Abelard's thoughts will be embedded in the larger milieu in which he studied and lectured.

Scholastic reflections on artifacts draw upon Aristotle's major physical and metaphysical works, including most importantly, his *Physics* and *Metaphysics*. Abelard and others in his milieu, namely, the loose circle of lecturers on logic and grammar centered around Paris, had a different set of authoritative texts on hand [1]. The only widely available Aristotelian texts were the *Categories* and *On Interpretation*. Supplementing this was Porphyry's *Isagoge* (or *Introduction*), Boethius's extensive commentaries on these works (including two full commentaries on the *Isagoge*), a partial translation of Plato's *Timaeus* with an accompanying commentary by Calcidius, and a handful of other introductions to logical topics by Boethius and other late Roman authors. For a general overview of Abelard's philosophy and his intellectual milieu, see [2,3] and the articles in [4]. There is presently no comprehensive study of twelfth-century philosophy, as it was practiced in Abelard's day, which is no doubt

due in part to the fact that much of it is preserved in anonymous treatises and records of lectures. However, one can consult the comprehensive study of the school of William of Champeaux in [5] and the introductory surveys in [6].

One might view this list of resources as impoverished, and surely the fact that certain key Aristotelian principles were not firmly in place in this period would explain to some degree the fact that much of the philosophy from the twelfth century was quickly sidelined in later discussions of physics and metaphysics. However, arguably this difference in resources is liberating. As I will show below, it leads Abelard and other twelfth-century thinkers down some interesting paths.

My tour through twelfth-century theorization about socially constructed entities will proceed as follows. In Section 2, I will briefly outline Abelard's theory of the creation and destruction of substances as it is presented in his synthetic masterpiece on logic, the *Dialectica* [7]. This will provide us with our first glimpse of how Abelard and others in his milieu think about the divide between substances and objects made by humans. Here I am careful to insist on the word "object", since it will become clear that for Abelard a "thing" (res)—that is, a part of reality (de rerum natura)—is something that God makes, not humans. In Section 3, I will examine Abelard and others on the identity and persistence of houses. Houses, Abelard claims, are interestingly analogous to substances in ways that other kinds of socially constructed objects are not. In Section 4, I will briefly consider the possibility that other objects to which we make reference, while anchored in the natural order of things in some sense, are nevertheless convenient fictions. Arguably, then, such objects as days and years, and perhaps also numbers, are artifacts in an important sense. Incidentally, it is here that Abelard and other twelfth-century philosophers echo themes and puzzles from even earlier in the Greek philosophical tradition (although, no direct connection has presently been established). Just like Democritean philosophers and the so-called Sophists of the fifth-century BCE, twelfth-century philosophers were deeply interested in questions about the "fit" between language and reality and where precisely the divide between phusis (or the realm of res) and nomos (or convention) is [8] (pp. 368–371).

## 2. Divine Creation, Human Making

A good entry into Abelard's understanding of the divide between nature and human activities and conventions is his reflection on the distinction that Aristotle makes in chapter 14 of the *Categories*, between substantial change and accidental changes, including, most crucially, the distinction between substantial change and change in quantity [9] (p. 15a).

Abelard notes that substantial change comes in two forms, generation and corruption. Generation occurs when something "takes on a substantial being" (substantiale esse assumit), as when a body (corpus) becomes a living body (corpus animatum) [7] (p. 418). Corruption is when a persisting thing leaves a substantial nature, as when this body ceases to be a living body and becomes an inanimate body. It is important to stress from the start that the body here is a persisting, actual substance. It is a thing with substantial forms that make it a corporeal substance. This particular corporeal substance takes on or leaves various states of substantial being by acquiring or losing various substantial forms. Thus, Abelard assumes the doctrine that there is a plurality of substantial forms in many concrete actual substances (see [7] (p. 419)).

Abelard is emphatic that generation is a process that only God, in his sole capacity as Creator, can bring about [7] (p. 418 and p. 419). In fact, God creates in two specific ways, both implied by Genesis 1:1. In "first creation", God makes both the substantial forms and the very "material" (materia) out of which all other things are constructed. These materials are the four elements. First creation, then, is creation ex nihilo. In "subsequent", or "second", creations, God joins substantial forms to materials in order to make the substances that fall into the very kinds described by traditional Aristotelians, including humans, horses, and also corpses. Abelard adds that it is in these secondary acts of creation that we seem to have generation and corruption in the strict sense: "In such a case, there is clearly no new material, but only a difference of form, and in so far as this involves a form

of a substance, we see a nature of a substance changed" [7] (p. 419). Abelard suggests that properly speaking Aristotle's notions of generation and corruption apply to the subsequent acts of creation, as these changes continue to happen as the world's history proceeds.

Abelard considers an objection to his dictum that only God generates [7] (p. 418): While we cannot generate substances, it certainly seems that we can *corrupt* them. We can, for instance, kill a man, burn wood down into ash, or melt sand into glass. Given that every corruption seems to entail a correlative generation, it also, therefore, seems that we humans can generate. We can make inanimate bodies, ash, and glass.

Abelard agrees that we clearly have a knack for "fickly" killing, and, more generally, for destroying the goods that God so generously creates. However, interestingly, all he concedes is that we can create *conditions* under which it is difficult for a substance to retain its decisive substantial form. By cutting off a man's head, we make it difficult for the body to continue to take on the substantial being of living rational body. What we do *not* do is actually join another substantial form to the body that has lost its rational form. As Abelard puts it, we can make it so that a body ceases to be animate, but we are not the ones who make the body inanimate [7] (p. 420).

Thus, according to Abelard, *all* substantial changes are due to God and God alone. We can assist by making the conditions right. The artisan places the sand in the furnace, whereupon God acts on the sand and converts it into glass. The father and mother come together and put the right materials into close proximity, but it is God who makes the human offspring in the womb [7] (pp. 419–20).

In the *Categories*, as Abelard and others read it, Aristotle divides change into four basic kinds:

1.  substantial change (generation and corruption);
2.  change in quantity (increase and decrease);
3.  alteration (change in quality);
4.  change in place.

We have now seen that humans are incapable of enacting substantial changes, but that we can aid in the manifestation of such changes by enacting changes in the placement of bits of the furniture of the universe. As it happens, it may be the case that the *only* thing we are capable of doing is making changes of the fourth type. To see this, consider how Abelard proceeds to analyze changes in quantity and to solve the riddle of whether anything grows.

Abelard's analysis of quantitative change tends to focus on the phenomenon of "increase" (augere), of which one species is the kind of increase that occurs when an animal "grows" (crescere). Yet, it is clear that his analyses and arguments are meant to apply mutatis mutandis to the correlative phenomenon of decrease. "A thing", he tells us, "is increased when the quantity of the substance increases through the addition of something" [7] (p. 421). Notice immediately how increase is tracking an accidental feature of a substance, which again, involves God's creative intercession. For when something is added to adolescent Socrates, it is God who "secretly ministers" to the expansion of Socrates along this or that dimension, whether it be his height or his girth. Likewise, Abelard insists that it is God who "attenuates our substance" by slowly subtracting "something" (aliquid) as we age.

Clearly Abelard is thinking about metabolism, which involves the incorporation of material into a substance. So, when I eat or trim my nails, all I am doing is providing the conditions under which God will act either to incorporate something into my substance or transform a clipping into something non-human. I seem to do this by moving bits of firstly created material around in space.

To see more clearly how human actions seem to involve merely moving bits of the universe around in space, consider how Abelard solves the puzzle of whether anything increases in the *Dialectica* (the treatment of the puzzle in the *Dialectica* is only one of several discussions of the problem [10–12]).

It appears that nothing increases or decreases. For increase seems to occur when something, call it X, is increased in some way by the addition of something else, Y. However, now consider X and Y prior to and then after the alleged change. At the prior time, both X and Y exist, X is a certain size S1, and Y is S2. If we then add Y to X by bringing them close together, notice that *X itself* is still S1, and Y *itself* is still S2. In other words, by adding Y to X, we have done nothing to *X*'s quantity. The only thing we have done is to move X and Y closer together. Now, as Abelard observes, we might think that this is what increase is, just as we might say that we have created an increase in men when we invite Plato into the house where Socrates is already sitting. However, Abelard hastens to assert that this is not what the "authorities" (viz. Aristotle and his commentators) have in mind. Nor is it in conformity with how we typically use the term "increase" [7] (p. 423). To preserve true increase (and decrease) we need to find a way to have it such that when something is added to X, *X itself* increases in dimension.

It is for this reason that Abelard concedes in the case of discrete quantities, especially numbers, that there may be a sense in which they do not increase or decrease [7] (p. 423). For the twosome of Socrates and Plato merely requires that Socrates and Plato both exist; they do not need to be close to one another. However, *continuous* quantities can be said to increase and decrease. In this case, Abelard argues, the key is that the material bits have been conjoined in the right way, so that the bits now conjoined are governed by a single form.

### 3. Survival Conditions for Houses

It is now time to consider some of the implications of Abelard's treatments of substantial and quantitative change. We can consider the issues under these heads: First, since it appears that increase and decrease can only occur for substances, what does this imply about the ontological stability of complex artifacts such as houses? Second, if there are analogies between houses and substances, does this erode the divide between substance and artifact that Abelard has tried to erect? Finally, what does the analysis of increase and decrease imply about human powers to effect changes in the world?

*3.1. Stability of Substances*

In the stretch of the *Dialectica* where Abelard explicitly discusses the problem of increase, it appears that increase and decrease can only happen to substances, since it is only here that we can get it so that some *X itself* can become bigger or smaller. At several places in his logical works, including in the course of his examination of increase, Abelard asserts that "no thing (res) has more or fewer parts at one time than at another". In addition, it is for this reason that he immediately concludes that "in a case where more parts have come to be when a substance has been changed in quantity, the substance itself cannot be said to have been that substance which had fewer parts" [7] (p. 423). The key to understanding what is going on here is to see that Abelard is equivocating between two senses of "whole" and "part", since he does acknowledge that a substance's "whole", and, hence, its quantitative parts, can increase. This is the whole, which he later calls an "integral" whole, that tracks the "ambit" of a substance (comprehensionem substantiae) [7] (p. 546). However, if we are thinking about increases in substance as such, this can only occur when God intervenes:

> But it has been said that two kinds of quantitative change can come to pass, since some of these changes occur with respect to the ambit of a substance, and then there are some that involve both, that is, both substance and quantity. For when substance is increased, it is also necessary that the quantity of substance grows. And this [latter] quantitative change has been relegated to [the kind of change] differentiated as the superior, that is, to the one that has been called coming to be substantially. For in a superior change, it is not necessary that the quantity of substance change, only the substance itself. For when the body of a man is changed from animate to inanimate, the quantity of substance is not changed, so

long as the members remain the same. But through an addition or subtraction of something it is necessary that the quantity, that is, the ambit of the substance changes [7] (p. 424).

There is a noticeable tension at work here in Abelard's treatment of increase and decrease. How seriously should we take his assertion that no res can have more or fewer parts at any given time? Is he really saying, as he seems to be, that a single substance can be different things at different times? After all, as we noted previously, Abelard seems to be committed to the notion that the material of the universe, once it exists, is merely moved about and imbued with various substantial forms for certain limited spans of time.

Perhaps we merely have a case here of Abelard asserting the rule about res and its mereological inflexibility for rhetorical purposes, specifically, in order to motivate the puzzle. One reason, however, to be skeptical of this idea is that later, when thinking about whether things can gain or lose parts, Abelard asserts that strictly speaking *this* house cannot persist if even one part is gained or lost. There might be a collection of bits and pieces on the lot that still have the form and function of a house after Rocky the stone is removed, but it is not *this* house, the one consisting of all the bits and pieces plus Rocky [7] (p. 550). Abelard resists this conclusion in the case of a substance, like Socrates. However, his reasoning is not fully filled out, and he seems to be willing to concede that Socrates or any other particular substance can perhaps consist of different things—that is, different collections of substantial material—at different moments in time: "Therefore, it does not appear to be the case that this man consists of all his parts taken together, but only of those apart from which he cannot be found" (p. 552). What is clear is how careful Abelard is to frame the problem of increase as a puzzle about the *quantities* of things. He seems to have thought that this thesis is compatible with the view that a substance might be identical with different collections of res governed by the final substantial form over the course of its lifetime.

However, can the center hold? More specifically, can Abelard carve out this position in such a way that substances and artifacts neatly fall on separate sides of a clearly demarcated line?

One way to test Abelard's position is to approach it from the side of substances. Can the mereological flexibility of substances hold despite the position that certain substances, the products of first creation, appear to be mereologically inflexible? Certain philosophers who are sometimes associated with Abelard and who have subsequently been named "Nominalists" (Nominales), for instance, reportedly maintained the thesis that "nothing grows" (nihil crescit) [12]. Perhaps the most striking expression of this thesis can be found in an anonymous commentary on the *Categories*:

This rule seems to make it possible to lodge a potential difficulty for us, as we say that every removal of a part, every addition, and every transposition of a parts changes the being of the whole (essentiam totius). Yet that which is sick cannot cease to be sick unless either some part is subtracted, or some part is added, or some of the parts are transposed. Therefore, a case of being sick cannot cease unless being ceases. And hence, there cannot be a being healthy [13] (p. 397).

Here, we seem at first to have precisely the sort of collapse of substances into specific sets of things governed by a dominant substantial form that many later nominalist thinkers are attracted to. For instance, in the fourteenth-century philosophers such as Jean Buridan maintained that natural substantial forms cannot transfer from one set of material bits to another. Hence, if Browny the horse's material bits change, Browny's horse form is also different. "Browny", consequently, turns out to be a substantive noun that behaves more or less in the same way that "the Seine" behaves. It is a noun that picks out different substances at different times, where the governing pragmatic principle at work is the fact that the sets of substances involved are appropriately connected to one another by means of some natural, causal story (for more on Buridan and Browny, see [14–17]).

It is interesting, therefore, to see that this particular "Nominalist" does not go down that path: "To this we say that whatever is healthy is by its nature healthy, and whatever is sick is by its nature sick; nevertheless, *the same man* who is healthy can be sick" [13] (p. 397, *my emphasis*). In other words, this Nominalist seems to adopt the picture that we are here attributing to Abelard. This is not to say that there were not any in Abelard's day who maintained this harder position about the tenability of substances. However, if we are looking for a clear example of a work by an early Nominalist that anticipates Buridan's position, it must be found elsewhere in the mass of manuscripts still waiting for proper, thorough study.

*3.2. Analogy of House Forms to Substantial Forms*

So far, then, it appears that Abelard and perhaps most twelfth-century thinkers maintain that substances are different from collections of discrete things, in virtue of the fact that while collections of discrete things are determined by the precise components that comprise the collection, substances can gain or lose members of the collection of materials that underlie them, so long as the dominant substantial form is still holding the remaining bits together into a continuous unity.

As mentioned above, it seems that at first Abelard wishes to enforce the divide by appealing to the continuous nature of substances. That is, because the form unifies the substance, its quantitative parts are continuous with one another. For Abelard as well as for many others in this period, artifacts are, by their very nature, things that have discrete parts. Indeed, for some the lack of continuity between the parts of an artifact immediately guarantees that the artifact is somehow less than fully one. Here, for instance, is an anonymous author from the school of Gilbert of Poitiers.

> The reason why it is claimed that every contiguous whole is many, that is, [identical] to its parts, but that no continuous whole is: [ . . . ] Among integral wholes, there are some that are aggregated out of their parts, as a tree is, and there are some that are disaggregated, as a flock is. Of the aggregated wholes, some are continuous (like a stone), others are contiguous (like a corset), and others are successive (like time). Therefore, a continuous whole is one whose parts are naturally conjoined to one another in such a way that they are believed to contain one another or at least to hold themselves together. For this reason the [parts] are said to be continuous, as they are in a stone, for which it is truly said that it is one. But a contiguous whole is one whose parts are naturally disjoint, while being conjoined by means of art, as one sees in a corset or a woven fabric. Here a part is placed next to a part and aggregated so as to make a whole. But since there is no difference between the whole called contiguous and the whole called disaggregated or collected save that the parts of the former are closer to together, whereas those of the latter are dispersed either by art or by chance, what reason can there be why the one that is disaggregated is many, yet the [whole] that is contiguous is one? [18] (pp. 38–39).

Since a house is merely contiguous, and not truly continuous, it is identical to its parts, and, accordingly, if a part is removed or added, it in the strict sense is no longer there. As we have seen, Abelard seems to endorse this line of reasoning in his discussion of the principal parts of things [7] (pp. 549–52).

There is a complication, however, that Abelard himself initiates. Recall Abelard's account of what increase is. It is not a case where we judge that either X or Y, or even the collection of X and Y, is bigger. Prior to being adjoined to one another, X and Y both existed, were each the size that they still are, and, hence, the collection of the two already existed and is the size that it is. Rather, Abelard argues, it is more appropriate to say that prior to joining X and Y, X and Y did not form a composite.

> Hence, when we say 'if something (quid) is added to some thing (cuilibet rei), the whole is made bigger', this should not be taken to mean that the composite has

become bigger than it was before. Rather, a composite that is bigger than each of the parts taken by themselves has been made by an act of joining some of them together. For in fact previously there was no composite [7] (p. 423).

Later in his *Dialectica*, Abelard observes that there are several grades of integral whole [7] (p. 548). Some integral wholes are mere pluralities. In such cases, the parts X, Y, and Z need not even be spatially proximate, let alone arranged in any meaningful way, in order to be a whole. Others, such as crowds and flocks, are aggregated wholes. In these cases, X, Y, and Z must be spatially proximate to one another for a whole of this sort to exist. However, there is a third sort of whole, whose parts must be more than merely spatially proximate to one another. In these cases, the parts must also possess a "composition" (compositio). Here, Abelard means that the parts must be organized in a specific way. This organization can be due to a substantial form, in which case the parts are united under a single nature. However—and this is the crucial point—there are also wholes that are unified in an important sense, but whose form is accidental in nature. Houses and dresses are such items. For, as Abelard observes, it is not enough to have all the house's parts on the lot. They must be composed in order to complete a house [7] (pp. 550–551). In one of his discussions of grades of wholes, Abelard observes that there is a difference in how nouns referring to these objects work [19] (pp. 170–171). Mere pluralities of discrete particulars take plural nouns. However, organized wholes, including ships and houses, behave syntactically like names for particular substances. That is, the nouns are used in their singular form, even if ontologically the house is many.

It is, perhaps, in part due to this fact about how Latin works that Abelard is tempted to muse about the analogy between the composition of a human and the composition of a house, when he turns from a discussion of integral wholes (which, remember, are tracking the "ambit"—that is, quantity—of a substance) to a discussion of wholes understood "in terms of their substance and their form" (secundum substantiam et formam) [7] (pp. 559 f.).

Abelard makes it plain that the mereology at work in hylomorphic composition is different from aggregation. A hunk of material, M, and a substantial form, F, must be "conjoined" (coniungere) in order to have an instance of a substance belonging to some substantial kind (the F-en things). However, the substance that arises is not merely an aggregate of M and F. Rather, it is through the conjunction of the two that M "transitions into" a new nature. This allows us to then say that M conjoined to F is nothing other than the item that is materialized (ipsum materietatum).

It is at this point that Abelard observes a parallel between substantial hylomorphic conjunction and accidental hylomorphic conjunction. Statues are not substantial, as the authorities make abundantly clear. Nevertheless, there is a likeness between statues and humans, which suggests that the composition that inheres in the statue is "quasi-substantial" [7] (p. 561). Specifically, there seem to be two analogies. First, a statue cannot be without its composition, any more than Socrates can be without his rational substantial form. Second, like Socrates, a statue is nothing other than the material that has been arranged in the right way. The statue, in a sense, is nothing other than the bronze shaped in a certain manner. A golden ring is "nothing other than the gold once it has been forged into a round shape", and "this house is nothing other than this wood and these stones to which this compositio has been superadded" [7] (pp. 560–561).

Again, Abelard interprets these claims about artifacts in such a way that the house being nothing other than these boards and these stones implies that each of the boards and stones is necessary in order to have this house. However, interestingly, Abelard is challenged on this point by many of his contemporaries. Two exhibits from the twelfth-century milieu stick out. The first is a short fragment on the persistence of houses that has been edited by Peter King as part of a longer treatise on universals, which he believes to be from a follower of one Joscelin of Soissons [20] (p. 105). The second is an introduction to logic from the school of one of Abelard's most astute critics, Alberic [21].

The discussion in the Joscelinian fragment starts with the startling observation that we could think of the house as *either* a continuous whole *or* as a discrete whole [20] (p. 122). It

is not clear by this remark whether the author seriously believes that houses are continuous wholes. Perhaps instead, this is merely a ploy. For the author proceeds to demonstrate that, in either case, this house can persist as this house even if it were to lose a stone—nay, even half a wall—provided that the half of the wall that is removed does not make it such that the remainder cannot go on as a house. That is, a house can lose a significant portion of one of its walls, provided that this portion is only a quantitative part of the wall. As long as the wall's substance, and, thus, its capacity to be a wall for this house, is retained, the house is retained (pp. 124–128) (see also the study of these passages in [22]).

The discussion and even some of the solutions mooted in the Joscelinian fragment echo the discussion of the house and its persistence conditions in the *Introductiones Montanae Maiores*. In particular, the author of the *Introductiones* observes that the alleged topical rule, that if a part is destroyed, the whole is destroyed, must be restricted to principal parts [21] (p. 132). The principal parts may be either substantially principal parts, as the heart and brain are for a human, or quantitatively principal parts, as the walls, roof, and foundation are for a house. This suggests that, unlike the Joscelinian work, the author of the *Introductiones* is under no illusion that a house might be a true continuous whole.

The Joscelinian author concedes that if a house were a discrete whole, it would depend upon *its* parts. However, he argues that it does not follow that any removal of any part *of a part* of the house entails the destruction of the house as a whole, because not every removal of parts of parts entails the removal of the house's part. The example that the author gives is of a flock of sheep [20] (p. 126). This flock must have, say, A, B, and C. However, if one were to shave A, this would not, thereby, destroy the flock consisting of A, B, and C. Whereas if one were to remove A's heart, this would destroy this flock, even though another flock might persist. Similarly, some of the parts of a house might be true continua, in which case a reduction in quantity of any of these parts would not entail the destruction of the whole house.

The author of the *Introductiones*, by contrast, seems to offer up a much more radical proposal. To say that a wall is a principal part of a house means that houses must have walls in order to be houses. However, it does not entail that this house must have this wall [21] (pp. 132–33). Likewise, it is true to say that when one of the smallest parts of a house is destroyed, the whole house is destroyed, so long as we interpret this to mean that if there were no smallest parts, there would be no house (p. 134). Again, however, none of this entails that this house must have this determinate little stone. If a specific house, consisting of a set of parts P and a stone S, were to lose S, this is all that is true: the house "does not continue on in the whole's same distinctive feature" (non remanet in eadem proprietate totius) (p. 134). It is hard to render the notion into eloquent English, but the basic idea is clear enough (remember that for an Aristotelian in this period, a proprium or proprietas is a distinctive or characteristic feature of a thing or kind of thing). If the house loses S, the house loses the distinctive features of a certain whole. In particular, the house is no longer *a whole consisting of P and S*. However, this does not entail that the *house* ceases to be. The author of the *Introductiones* then adds this especially intriguing principle of determining whether "the F" continues to exist: The house is said to continue on when some parts of it remain, just as the king is said to continue on in his sons, even after he has perished, when some one of his sons lives and continues on father's role as king (pp. 134–135).

### 3.3. Humans and Their Impact on Reality

The example of *the* king persisting through a series of humans is intriguing because it suggests that for the author of the *Introductiones*, artifacts are intimately tied up with human activities and conventions. Not only is it the case that their existence is initiated by humans, but they *persist* so long as humans treat certain things in a specific vicinity and with the right sorts of connections, such as having certain properties or playing certain roles. In short, for this author, the activities of humans may have an impact of the furniture of the world. Some of the items that exist are socially constructed entities.

If that is right—and, admittedly, this is a big "if"—we can begin to see an important contrast between the Alberici and Abelard. For, while Abelard seems to be willing to allow us to talk about houses as if they are substances, it still seems true to say that when we are looking at the actual furniture of the world, at the bottom there is material and material unified under some substantial form. On top of that, there are various accidental arrangements and configurations of substances. However, our impact on what "really" exists is minimal. We can provide or remove *conditions* for the being and persistence of res, but we have no agency in making additional res. More to point, when we take some res and manipulate them so that they take on the composition of a house, Abelard is very clear in indicating that what we have not done is make any additional res.

## 4. Are Artifacts Convenient Fictions?

What, then, are we doing when we manipulate and reorganize the world of res? Here, we must move cautiously and tentatively, but one way to possibly construe what Abelard is up to is to compare houses and statues to his analysis of items, which he unambiguously identifies as convenient fictions.

Abelard's treatment of the persistence of houses and humans appears in his discussion of integral wholes, which are intimately linked to the quantities of substances. After settling the question of which, if any, parts of integral wholes are principal, Abelard turns to consider another set of quantities that are treated as wholes. These alleged integral wholes, however, are different from the others. In particular, they seem to behave in ways *opposite* to those of most other integral wholes. Other integral wholes may be able to gain or lose some of their parts, but, nevertheless, it must be the case that there are always several parts present for the whole to be present. Successive wholes, including temporal wholes (the central focus in the *Dialectica*) but also uttered speeches, on the other hand, seem to be incapable of having more than one part present at any given point in the whole's existence. When the first hour of the day is, no other hour yet exists. When the second hour exists, the first is no more and the rest are yet to come. The same is true for a spoken statement, such as "I am a human". When "an" is uttered, not only the "I", the "am", and the "a", but also the "hum" have passed away into nothingness.

There are two options at this point: either (i) one can admit that there are some wholes out there in the world whose very nature requires that they always consist of only one part at any given moment, or (ii) one can insist that these "wholes" are not part of reality as such, but are constructions out of bits of reality that we hold in our mind and talk about because of some utility that they offer. Abelard unambiguously goes for the second of these alternatives: "Therefore, if we are to insist on the truth of the matter, we must admit that these [temporal items] are not wholes, but that they have been treated as if they are wholes by philosophers, namely, on account of the fact that, so that they might point out their nature they collected those that were previously or will be with that which presently is in their thought as if they are one and as if there were something that consists of what is with what is not" [7] (p. 554). Days are no more parts of reality than is a whole consisting of a human and a chimera.

It is important to understand that Abelard does not think that items such as days have no basis or anchor in reality. When we think about a day, we are tracking natural states of affairs and processes. These processes even behave according to fixed rules. In order to think about a day, we have to hold in our minds a set of thoughts about actual states of affairs that proceed successively one upon the other in the right way. The same is true of an instance of spoken words. Not just any assemblage of sounds held together all at once in the mind will constitute a meaningful quasi whole.

Nonetheless, what is true is that the world has within it fewer assemblages of things than our language or our ways of thinking present to us. Just how radically austere is this picture? Abelard clearly maintains a fictionalist position with respect to successive wholes: There are things in certain states, but there are not in addition successive things composed out of these simpler. Abelard is a little harder to pin down when it comes to whether there

actually are, independent of our human purposes and projections, any wholes consisting of simultaneously existing yet entirely discrete objects. At some points, it seems that he is willing to concede that these are real (see [7] (p. 547)).

However, as Abelard notes, there are some in his day who did wonder whether *any* collection of items that are not united into one single nature—that is to say, whether *any* alleged item that is not continuous—can be part of the natural order of things (in natura rerum). Some in Abelard's day argued explicitly that all numbers, both two itself but also concrete pairs, are only treated *as though* they are one thing by humans (see [5] (p. 363), with discussion in [5] (pp. 157–165) and [23]). We have also seen in the Porretan logical treatise that there is no difference between a number and a house in terms of whether they are many as opposed to one. Putting these two thoughts together would imply that all discrete wholes are merely fictions; they are conglomerations of things that we treat *as if* they are one thing. However, did anyone actually make such a leap?

Perhaps so: at the end of his treatment of integral wholes, Abelard pointedly criticizes his old teacher Roscelin. Roscelin, he claims, had the "insane" view that "no thing consists of parts" and, just as he said about universals, he asserts that all parts are "merely words" [7] (p. 555). He then presents what he alleges are two arguments that Roscelin used to defend this claim, immediately followed by forceful rebuttals. There are a host of questions about this section of the *Dialectica*. Is Abelard faithfully presenting Roscelin's arguments or is he developing a strawman to knock down? How are the arguments supposed to work exactly? For one noteworthy attempt to find reason within Roscelin's "madness", see [24] (pp. 114–128). However, it is perhaps significant that the test case Abelard presents for the claim that parts are merely verbal is a house.

Now, strictly speaking, the dispute between Roscelin and Abelard is about whether any object can be divisible into real items. So, it is possible that Roscelin maintained that there are houses, but that these objects are not divisible into things. Yet, I think it is likely that Roscelin would have maintained what seems to have been a commonplace, namely, that houses are discrete wholes (remember: the only evidence that anyone denied this is found in the Joscelinian treatment of houses [20], which we have already noted is hard to interpret). If Roscelin believed this, then perhaps he also believed that houses are merely conglomerations of things that we speak of as if they were one item. Here are some reasons to entertain this speculation. Roscelin almost surely would have held, like Abelard, that "part" and "whole" are correlative notions. Thus, if Roscelin maintained that "parts" are merely verbal, it would seem that he must also hold that "wholes", which are composites of parts, are merely verbal. After all, as we have already seen Abelard assert, no true whole can consist of merely one thing [7] (p. 554). Thus, Roscelin could have maintained that there are things, each of them utterly unitary, but denied that any of these things made an additional thing by entering into a composition relation (this, in effect, is the interpretation that Kluge has offered [25] (pp. 409–412)). It is only a short step from here to the notion that not only house parts but also houses are merely verbal, especially if, like Abelard, Roscelin maintained that houses are nothing other than stones and boards arranged in a house-like manner.

All of this is admittedly speculative. However, it may have been the case that Roscelin and perhaps others from Abelard's milieu actually inhabited the logical space that appears when we combine the notion that no discrete collection is naturally unified with the idea that any item that is not naturally unified is only treated *as if* it were a whole by humans in order to serve some practical purpose.

Abelard was not willing to follow Roscelin's position about parts (universals, as is well known, are a different story). This might also mean that Abelard is not willing to go so far as to assert that houses are merely convenient fictions, even if they are manmade and, hence, not themselves res or substances.

## 5. Conclusions

I have argued that Abelard tries to maintain a sharp distinction between things (res) and substances, on the one hand, and the products of human art, language, and thought, on the other. Admittedly, it is not entirely clear just how extreme Abelard's position is. It does appear that Abelard attributes to human agency only the capacity to produce conditions under which God creates, and that we only do this by moving bits of reality around in space. Yet, his rebellion from his old teacher Roscelin suggests that he might not have been willing to go all the way to an extreme fictionalist position about artifacts such as houses.

We have also noticed some instances where Abelard acknowledges certain affinities between houses and statues and substances. Abelard's hesitancy in such places about the divide between things and artifacts might have been the impetus for a variety of competing theories of artifacts. Of particular note has been the thesis proposed by the followers of Alberic: that artifacts and other social constructions have persistence conditions that are strikingly similar to those of substances. This is not to say that the Alberici and like-minded folk were dispensing with Aristotelian categories altogether. A line is still drawn between the substantial and the artificial. However, this twelfth-century version of Aristotelian philosophy is what we might call a "lite" version of Aristotle because it neither enforces a sharp line between substance and artifact in the way that, say, Aquinas does, nor proposes the reading of Aristotle's physics that inspires later thinkers such as Ockham and Buridan. It is such a lite version that we find in the modern-day "neo-Aristotelianism" of such philosophers as Kathrin Koslicki [26] and Mark Johnston [27]. It is possible, then, that the moves that twelfth-century philosophers made will be of interest to modern-day philosophers, since many contemporary thinkers who find "forms" and "structures" appealing seem to be less enthusiastic about embracing the strict divide between substances and artifacts found in Aristotle's *Physics* and *Metaphysics*.

**Funding:** This research received no external funding.

**Institutional Review Board Statement:** Not applicable.

**Informed Consent Statement:** Not applicable.

**Data Availability Statement:** Not applicable.

**Acknowledgments:** My thinking about Abelard has changed considerably over the years, in no small part because of the published and unpublished work of Peter King and Christopher J. Martin. They helped me to see that my initial assessment of Abelard as a much more radical Nominalist is not warranted by the texts, especially once they have been properly edited.

**Conflicts of Interest:** The author declares no conflict of interest.

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
