# Peer review of "Abelard and Other Twelfth-Century Thinkers on Social Constructions"

_philosophies, doi:10.3390/philosophies7040084_

Round 1
Reviewer 1 Report
My only suggestion is for the author to relegate the discussion of Haslanger to a footnote because this is largely a work in medieval philosophy and because it doesn't purport to be a contribution to the contemporary literature in social ontology. The off-handed comments referring to Haslanger, as well as Koslicki and Johnston, actually take away from the overall paper.
Author Response
I have deleted the reference to Haslanger, as I agree with the referee's point about that digression.
I have kept the references to Koslicki and Johnston in the conclusion, but I have modified the language so that it is clear that it is only a tentative suggestion. I agree that the paper is primarily historical, but it would be a bonus if others outside of our specialist field found something useful in this special issue (as I think some of them might).
It should be noted that the other referee asked that I add a reference to the Sophists and that I add more bibliography about Abelard and twelfth-century philosophy. I tried to do this in as unobtrusive a manner as possible.
Reviewer 2 Report
The topic is extremely interesting, Abelard is a very inspiring philosopher. The abstract is concise. The original dispute, which Abelard commented on, is of ancient origin. The sophists mention it. It would be appropriate for the author to point out the original distinction among sophists between what is natural and what has been introduced by man. The professional level of the text is good. It can be seen that the author is familiar with the issue of the availability of Aristotle's texts in the 12th century, well before the central translations into Latin. The paper reveals interesting parts of Abelard's metaphysics. I evaluate the comparison of Aristotle's works available at the time and Abelard's texts in the issue of the studied difference positively. Abelard's reflections on extinction are very detailed, from a metaphysical point of view. Abelard strives to build on the theory of change from Categories. De Generatione et Corruptione did not appear to have been translated at that time. The author discusses the metaphysical solution of the problem in Abelard (Dialectica) in great detail, precisely. This also needs to be assessed positively. He pays attention to quantitative changes. Changes are happening in the area of ​​the substance. The issue of division into parts (res) has a much later interesting solution at Duns Scotus. After evaluating the solution at Abelard, the author moves on to solutions with other philosophers. Buridan also got on the market. The author also compares House Forms and Substantial Forms between Abelard and Gilbert of Poitiers. The author's analysis is detailed and useful due to the genesis of disputes of realism and nominalism, as well as the later emergence of formalitates. The author took over the question of the relationship between form, whole and part in Abelard very well. The paper is an excellent probe into Abelard's metaphysical thinking. We must praise the author for touching even the little-known protagonist of the discussion of nominalism and realism as Joscelin of Soissons. The paper also seeks to capture the contrast between Abelard and Alberici. Later, similar language problems are solved in a different way. An analysis of Abcelrd's reflection on Roscelin is also valuable, as direct texts have not been preserved from him. The conclusion is quite extensive.
I recommend making small additions first. To point out with at least a few sentences the original perception of the difference between what is from nature and what is from man in sophists. We also propose to supplement the reference literature with at least a few reference studies devoted to Abelard.
Author Response
A short reference to the Sophists and other presocratics has been added to the Introduction in a place that seemed fitting.
More bibliography on Abelard and 12th century philosophy has been added, again, in a manner that seemed best given the idiosyncratic formatting and reference system of this journal.